# A Two-Photon Fluorescent Probe for the Visual Detection of Peroxynitrite in Living Cells and Zebrafish

**DOI:** 10.3390/molecules27154858

**Published:** 2022-07-29

**Authors:** Zhencai Xu, Jin Qian, Yufeng Ge, Yalan Wang, Hongwei Chen

**Affiliations:** 1Guanyun People’s Hospital, Lianyungang 222000, China; zhaoll0326@163.com; 2Modern Education Technology Center, Department of Critical Care Medicine, The First Affiliated Hospital of Hainan Medical University, Hainan Medical University, Haikou 571199, China; jang9116@hainmc.edu.cn; 3Department of Emergent Trauma Surgery, Qilu Hospital of Shandong University, Jinan 250012, China; geyufengqilu@163.com

**Keywords:** peroxynitrite, fluorescent probe, cell imaging, zebrafish

## Abstract

Peroxynitrite (ONOO^−^), as an important reactive oxygen species (ROS), holds great potential to react with a variety of biologically active substances, leading to the occurrence of various diseases such as cancer and neurodegenerative diseases. In this work, we developed a novel mitochondria-localized fluorescent probe, HDBT-ONOO^−^, which was designed as a mitochondria-targeting two-photon fluorescence probe based on 1,8-naphthylimide fluorophore and the reactive group of 4-(bromomethyl)-benzene boronic acid pinacol ester. More importantly, the probe exhibited good biocompatibility, sensitivity, and selectivity, enabling its successful application in imaging the generation of intracellular and extracellular ONOO^−^. Furthermore, exogenous and endogenous ONOO^−^ products in live zebrafish were visualized. It is greatly expected that the designed probe can serve as a useful imaging tool for clarifying the distribution and pathophysiological functions of ONOO^−^ in cells and zebrafish.

## 1. Introduction

Peroxynitrite (ONOO^−^), as representative active oxygen, shows high reactivity in living organisms [1,2,3]. It plays an important role in the physiological and pathological processes of living systems [4,5,6]. The transformation of ONOO^−^ in biological systems mainly involves two pathways: protonation to generate peroxynitrous acid (ONOOH), and quickly reacting with CO_2_ to form a short-lived intermediate nitrosoperoxycarbonate (ONOOCO_2_^−^) [7,8]. ONOO^−^ can diffuse freely through the phospholipid membrane bilayer, and its metabolites can react with a variety of important biomolecules (including proteins, lipids, and nucleic acids), eventually leading to mitochondrial dysfunction and cell death [9,10,11]. However, the overproduction of ONOO^−^ in vivo can lead to abnormalities in a variety of life targets, such as DNA, proteins, enzymes, and nucleic acids, which can, in turn, cause many diseases, such as cancer, Alzheimer’s disease, and nervous system degeneration [12,13,14,15]. Therefore, it is urgently needed to develop an accurate ONOO^−^ detection method that plays an important role in the indepth understanding of complex diseases in living systems.

Traditional methods of detecting ONOO^−^ are usually time-consuming and expensive [16,17,18]. ONOO^−^ also has some features such as instantaneity, low lifetime, low in vivo concentration that render its effective capture and further detection great challenges. Fluorescence imaging technology has developed rapidly in recent years. It possesses the advantages of high sensitivity, selectivity, in situ detection, and noninvasiveness, which have attracted the attention of an increasing number of researchers [19,20,21,22,23,24]. A variety of fluorescent probes for detecting ONOO^−^ have been developed and have had widespread applications [22,23,25,26,27,28], but there are few two-photon probes used to detect the distribution level of ONOO^−^ in subcellular organelles. Two-photon imaging itself has the characteristics of high resolution, high throughput, noninvasiveness, and excellent imaging depth [29,30,31,32], which enables it to exert the dynamical visualization functions of living organisms and cells in an active state, thereby facilitating researchers in exploring the changing levels of ONOO^−^ in cells and organisms in physiological and stimulating states.

On the basis of the above information, we developed a novel two-photon fluorescent probe, HDBT-ONOO^−^, for monitoring ONOO^−^ in mitochondria. HDBT-ONOO^−^ was equipped with cationic triphenylphosphine and borates groups as mitochondrial targeting groups and peroxynitrite responsive sites, respectively. Spectroscopic experiments confirmed the excellent selectivity and sensitivity of HDBT-ONOO^−^, and cell imaging experiments demonstrated that HDBT-ONOO^−^ showed mitochondria-targeting abilities and could be successfully used for two-photon imaging of ONOO^−^ variations in mitochondria in living cells. The following zebrafish experiments further proved that HDBT-ONOO^−^ had excellent sensitivity in the detection of the level of ONOO^−^ in the body. The above results confirm that HDBT-ONOO^−^ holds great application value, and lay the foundation for us to further study the pathophysiological processes with ONOO^−^ involved in vivo.

## 2. Experimental Section

### 2.1. General Comments

Details of materials and measurements were transferred to Appendix A.

### 2.2. Synthesis of HDBT-ONOO^−^

Intermediate compounds in Appendix A.

A mixture of compound 4 (258 mg, 0.5 mmol), 4-(bromomethyl)-benzene boronic acid pinacol ester (297 mg, 1 mmol) and anhydrous K_2_CO_3_ (138 mg, 1 mmol) in 10 mL of DMF was refluxed at 80 °C overnight. The solvent was removed, and the obtained crude product was purified with column chromatography using dichloromethane:methanol (50:1–20:1) to obtain a yellow solid HDBT-ONOO^−^. ^1^H NMR (500 MHz, CDCl_3_) δ 8.66–8.49 (m, 2H), 8.17 (s, 1H), 7.87 (dd, *J* = 33.0, 7.6 Hz, 3H), 7.73 (d, *J* = 10.4 Hz, 6H), 7.47 (ddd, *J* = 39.3, 26.7, 7.6 Hz, 11H), 7.07 (dd, *J* = 26.9, 8.1 Hz, 1H), 5.24 (s, 2H), 4.29 (t, *J* = 6.4 Hz, 2H), 2.42 (s, 2H), 2.08 (s, 2H), 1.37 (s, 12H). ^13^C NMR (126 MHz, CDCl_3_) δ 164.4, 160.7, 138.5, 135.3, 135.2, 135.1, 133.5, 131.7, 130.8, 130.8, 128.8, 128.7, 128.6, 127.4, 126.7, 126.1, 115.1, 106.5, 84.0, 83.9, 70.84 (s), 65.5 (s), 27.9 (s), 24.9. LC-HRMS (ESI, negative ion mode): *m*/*z* [C_32_H_22_N_3_O_5_P^+^], calcd,732.3049; found [M]: 732.3047.

### 2.3. Statistical Methods

The experimental data were analyzed using SPSS17. The 0 software package was used for statistical processing, measurement data are expressed as mean ± SD, and the *t*-test of two independent samples was used for comparison between groups. *p* < 0.05 was considered to be statistically significant.

## 3. Results and Discussion

### 3.1. Design of the Probe HDBT-ONOO^−^

The structure of HDBT-ONOO^−^ to ONOO^−^ and the proposed response mechanism are presented in Figure 1. As shown in Figure 1, the probe was designed by binding a 1,8-naphthylimide fluorophore scaffold modified with 4-bromomethylphenylboronic acid pinacol ester, where it exhibited bright fluorescence upon the conversion of borates into the corresponding phenol by ONOO^−^. The introduction of cationic triphenylphosphine enabled the probe to be highly localized near the subcellular organelle mitochondria [33,34,35]. The probe was employed for the sensitive and selective detection of both exogenous and endogenous ONOO^−^. The structural characterization of target substances was performed with ^1^H NMR, ^13^C NMR, and HR–MS (Appendix A).

### 3.2. Spectral Response of HDBT-ONOO^−^ to ONOO^−^

We explored the spectral properties of HDBT-ONOO^−^. The absorption and fluorescence emission spectra of the probes were first evaluated separately under simulated physiological conditions. As shown in Figure 1A, the addition of ONOO^−^ resulted in the gradual disappearance of the absorption band centered at 372 nm in HDBT-ONOO^−^, while a new red-shifted band appeared at around 450 nm. Next, we investigated the fluorescence spectra of HDBT-ONOO^−^. As shown in Figure 1B, the fluorescence intensity of the probe at 558 nm increased as ONOO^−^ concentration increased. In addition, we found a satisfactory linear response relationship between the fluorescence intensities of HDBT-ONOO^−^ and the concentrations of ONOO^−^ (Figure 1D). The linear fitting equation was F_558nm_ = 91.18243 [ONOO^−^] + 107.1069, and the correlation coefficient (R^2^) was 0.99687. On the basis of the standard method of 3σ/k, the detection limit of ONOO^−^ was calculated to be 56 nM. Later, we investigated whether pH could affect changes in probe fluorescence intensity. As shown in Figure 1C, HDBT-ONOO^−^ showed a weak fluorescence signal in the studied pH range (3.0–10.0). After adding a certain amount of ONOO^−^ (20 μM), the fluorescence intensity of the probe gradually increased with the increase in pH, and the fluorescence remained relatively stable in the pH range of 7–10. Subsequently, the fluorescence intensity gradually weakened with the further increase in alkalinity. These results indicate that HDBT-ONOO^−^ could be suitable for the detection of ONOO^−^ content under physiological conditions. The time course of HDBT-ONOO^−^ fluorescence emission at 558 nm after the addition of ONOO^−^ (20 μM) was next investigated. Appendix A shows that the fluorescence intensity of the probe increased with time and reached a maximum at around 30 s. To further confirm the specificity of the probe HDBT-ONOO^−^ for ONOO^−^, we tested the ability of the probe HDBT-ONOO^−^ to discriminate ONOO^−^ from other biologically relevant species, including metal cations (Na^+^, Ca^2+^, Mg^2+^, Zn^2+^, Fe^2+^, Al^3+^, Cu^2+^) and other ROS. As shown in Figure 2, only ONOO^−^ caused an observable fluorescence response, which indicated that HDBT-ONOO^−^ had excellent selectivity and selectivity to ONOO^−^ (Figure 2). These results indicate that HDBT-ONOO^−^ could be suitable for detecting ONOO^−^ content under physiological conditions.

### 3.3. Fluorescence Imaging in Living Cells

Encouraged by the above experiments, we then explored the potential of the probe in biological applications. As shown in Appendix A, HeLa, RAW 264.7, and HepG 2 cells maintained a high survival rate after being exposed to probe concentrations below 70 μM. SIN-1 is a well-known donor of ONOO^−^ [36]. As shown in Figure 3, as the concentration of SIN-1 increased, the fluorescence intensity of the probe gradually increased and reached a maximum with the concentration of SIN-1 at 1.2 m. Figure 3D shows that the fluorescence intensity of HDBT-ONOO^−^ was significantly attenuated after the addition of ONOO^−^ scavenger ebselen (200 µM). This revealed that our probe HDBT-ONOO^−^ could sensitively detect the changes in exogenous ONOO^−^ in cells.

To verify the reactivity of the probe to the intracellular ONOO^−^, the fluorescence imaging of RAW 264.7 cells was performed. Lipopolysaccharide (LPS) and interferon-γ (IFN-γ) could stimulate the production of ROS/RNS in RAW 264.7 cells to produce endogenous ONOO^−^ [37,38,39,40]. As shown in Figure 4, significant green fluorescence in the cytoplasm indicated that HDBT-ONOO^−^ could respond with ONOO^−^ in the cell. Then, ebselen was added in the presence of LPS and IFN-γ, and as expected, no changes in fluorescence were observed (Figure 4C). These results indicate that HDBT-ONOO^−^ can detect endogenous ONOO^−^ in living cells, so it has potential for imaging applications.

Lastly, using Mito-Tracker Deep Red (MT Deep Red), a commercially available mitochondrial dye, the subcellular distribution map of HDBT-ONOO^−^ was drawn through costaining experiments. As shown in Figure 5, the green signal generated by HDBT-ONOO^−^ in response to exogenous ONOO^−^ overlapped well with the red fluorescence of MT Deep Red. The calculated Pearson correlation coefficients were evaluated as 0.88, providing direct evidence that the probe could be selectively and effectively localized to the mitochondria of HepG 2 cells to detect slight ONOO^−^ level changes. To study the distribution of the probe in the cell, HDBT-ONOO^−^ was incubated with the ONOO donor SIN-1 or nuclear dye DAPI (4′,6-diamidino-2-phenylindole, which was a fluorescent dye that can bind strongly to DNA). As shown in Figure 6, in RAW 264.7 and HeLa cells, the fluorescence of the probe was mainly distributed in the cytoplasm, not in the nucleus.

### 3.4. Imaging of ONOO^−^ in Zebrafish

Using HDBT-ONOO^−^ as a probe, we demonstrated the fluorescence imaging of exogenous and endogenous ONOO^−^ generation in a zebrafish model. As shown in Figure 7, we observed only a weak fluorescence response in the control group. After the incubation of zebrafish with ONOO^−^, bright green fluorescence was observed. Similarly, upon stimulation by LPS, endogenous zebrafish ONOO^−^ was formed and thus exhibited intense fluorescence. These images demonstrate that HDBT-ONOO^−^ can sensitively image both exogenous and exogenous ONOO^−^ production in zebrafish.

## 4. Conclusions

In conclusion, we developed a novel two-photon fluorescent probe, HDBT-ONOO^−^, for sensitive detection of mitochondrial ONOO^−^ in living cells. It consisted of 1,8-naphthalimide fluorophore modified with a triphenylphosphonium targeting group and a boronate-based molecule switch. The probe demonstrated the desired properties of high selectivity, excellent water solubility, and physiological pH response, along with low cytotoxicity, enabling the tracking of mitochondrial ONOO^−^ in living cells. Thus, the probe might serve as a tool for probing the biological roles of mitochondrial ONOO^−^ and facilitating the mechanistic investigation of mitochondria-targeting anticancer agents. Lastly, zebrafish experiments implied the potential application of HDBT-ONOO^−^ in the studies of the ONOO^−^ roles in live organisms. Therefore, the probe may be promising as a tool for exploring the biological role of mitochondrial ONOO^−^, and to promote a deep understanding of the molecular events and mechanism of ONOO^−^ in the body.

## Data Availability

The data that support the findings of this study are available from the corresponding author upon reasonable request.

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
