# Peer review of "A Two-Photon Fluorescent Probe for the Visual Detection of Peroxynitrite in Living Cells and Zebrafish"

_molecules, 2022, doi:10.3390/molecules27154858_

Round 1
Reviewer 1 Report
The work is devoted to the development of a new two-photon fluorescent sensor for peroxynitrite ion in a biological media. The fluorophore used was 1,8-naphthylimide. The authors carefully characterized the sensing properties of the obtained compound and demonstrated its work in various cell types and in zebrafish. The manuscript can be published after minor corrections:
in vivo should be itallic (in vivo)
The authors should add a comparison of the key parameters of the sensor they obtained with those already described in the literature (the links are given by them in the introduction). Apparently, it makes sense to compare in table such parameters as LOD, emission maximum, absorption maximum, sensor type, etc.
Figure S4 should be moved to text, it's important.
In Supmat: For the HRMS method, the m/z values ​​(observed and calculated) for the brutto formula should be given in the description of the compounds, and not only the spectrum image itself.
Author Response
Dear Editor and Reviewers,
Thank you very much for your positive evaluation of this manuscript. According to your and the reviewer’s advice, the manuscript entitled “A two-photon fluorescent probe for visual detection of peroxynitrite in living cells and zebrafish” has been revised.
We highly appreciate the great efforts and significant guidance of the referees and your close study concerning our manuscript. These comments are of great importance for revising and improving our work. As you indicated, we have revised the manuscript seriously and carefully, as follows:
1) We have corrected some errors in the manuscript.
2) We have compared our probe with other published probes.
Sincerely yours,
Zhencai Xu
------------
Guanyun People’s Hospital, Lianyungang, Jiangsu, 222000, China
Zhencai Xu, Email: xu2454287273@163.com
------------
Response to the editor and reviewers’ comments
Independent Review Report, Reviewer1
EVALUATION
The work is devoted to the development of a new two-photon fluorescent sensor for peroxynitrite ions in a biological media. The fluorophore used was 1,8-naphthylimide. The authors carefully characterized the sensing properties of the obtained compound and demonstrated its work in various cell types and zebrafish. The manuscript can be published after minor corrections:
1. in vivo should be italic (in vivo)
R: Thank you for your suggestion. We have revised the relevant errors in the manuscript.
The authors should add a comparison of the key parameters of the sensor they obtained with those already described in the literature (the links are given by them in the introduction). Apparently, it makes sense to compare in table such parameters as LOD, emission maximum, absorption maximum, sensor type, etc.
R: Thanks for your suggestion. According to your suggestions, to help readers better understand the research progress of ONOO- probes, we compare the proposed ONOO- probes with previously reported probes. For example: response time, emission wavelength, and limit of detection. We have added it to the supporting information.
Table S1. Comparison of reported fluorescent probes and our proposed probe for ONOO-.
Probe |
LOD |
Emission wavelength |
Absorption wavelength |
Detection time |
Response type |
Ref |
145 nM |
712/680 nm |
602 nm |
3 min |
Ratiometric |
1 |
|
150 nM |
652/493 nm |
514 nm |
4 min |
Ratiometric |
2 |
|
2.5 μM |
540 nm |
482 nm |
30 min |
Off-On |
3 |
|
449 nM |
508 nm |
420 nm |
15 min |
Off-On |
4 |
|
150 nM |
530 nm |
510 nm |
20 min |
Off-On |
5 |
|
310 nM |
560 nm |
355 nm |
15 min |
Off-On |
6 |
|
120 nM |
558/454 nm |
345 nm |
20 min |
Ratiometric |
7 |
|
56 nM |
558 nm |
450 nm |
30 s |
Off-On |
This work |
- Zhang, J.; Zhen, X.; Zeng, J.; Pu, K., A Dual-modal Molecular Probe for Near-infrared Fluorescence and Photoacoustic Imaging of Peroxynitrite. Analytical Chemistry, 2018, 90, (15), 9301–9307.
- Hou, J. T.; Yang, J.; Li, K; Liao, Y. X.; Yu, K. K.; Yu, X. Q., A highly selective water-soluble optical probe for endogenous peroxynitrite. ChemComm, 2014, 50, 9947-9950.
- Kim, J. Y.; Park, J.; Lee, H.; Choi, Y.; Ki, Y., A boronate-based fluorescent probe for the selective detection of cellular peroxynitrite. ChemComm, 2014, 50, 9353-9356.
- Wang, Z.; Zhang, F.; Xiong, J, H.; Liu, Z, H., Investigations of drug-induced liver injury by a peroxynitrite activatable two-photon fluorescence probe. Spectrochimica Acta Part A: Molecular and Biomolecular Spectroscopy, 2021, 246, (5),118960.
- Xia, L, L.; Tong, Y.; Li, L, S.; Cui, M, Y.; Gu, Y, Q.; Wang, P., A selective fluorescent turn-on probe for imaging peroxynitrite in living cells and drug damaged liver tissues. Talanta, 2019, 9140, (19), 30657-5.
- Chen, L.; Cui, C, Y.; Chen, J, R.; Xia, L, L.; Deng, D, W.; Gu, Y, Q.; Wang, P., A novel highly selective fluorescent probe with new chalcone fluorophore for monitoring and imaging endogenous peroxynitrite in living cells and drug-damaged liver tissue. Talanta, 2021, 1, (215), 30657-5.
- Cui, J.; Zang, S, P.; Nie, H, L.; Shen, T, J.; Jing, J.; Zhang, X, L., An ICT-based Fluorescent Probe for Ratiometric Monitoring the Fluctuations of Peroxynitrite in Mitochondria. Sensors and Actuators B: Chemical, 2021, 1, (328) 129069.
- Figure S4 should be moved to text, it's important.
R: Thanks for your suggestion. We've moved Figure S4 into the text.
- In Supmat: For the HRMS method, the m/z values ​​(observed and calculated) for the brutto formula should be given in the description of the compounds, and not only the spectrum image itself.
R: Thanks for your suggestion, we have given the m/z values in the revised manuscript. 1H NMR (500 MHz, CDCl3) δ 8.66 – 8.49 (m, 2H), 8.17 (s, 1H), 7.87 (dd, J = 33.0, 7.6 Hz, 3H), 7.73 (d, J = 10.4 Hz, 6H), 7.47 (ddd, J = 39.3, 26.7, 7.6 Hz, 11H), 7.07 (dd, J = 26.9, 8.1 Hz, 1H), 5.24 (s, 2H), 4.29 (t, J = 6.4 Hz, 2H), 2.42 (s, 2H), 2.08 (s, 2H), 1.37 (s, 12H). 13C NMR (126 MHz, CDCl3) δ 164.4, 160.7, 138.5, 135.3, 135.2, 135.1, 133.5, 131.7, 130.8, 130.8, 128.8, 128.7, 128.6, 127.4, 126.7, 126.1, 115.1, 106.5, 84.0, 83.9, 70.84 (s), 65.5 (s), 27.9 (s), 24.9. LC-HRMS (ESI, negative ion mode): m/z [C32H22N3O5P+], calcd,732.3049; found [M]: 732.3047.

Reviewer 2 Report
In the manuscript, the authors claim to develop a novel ROS-oriented fluorescent probe which is specifically sensitive to ONOO- and can target mitochondria upon loading in the cells. Despite a variety of ROS detecting fluorescent probes available on the market now, new probes specific only to particular ROS of interest and targeting specific organelle presents a great interest to the research community.
The probe can freely permeate the cell's membrane and targets mitochondria due to the cationic triphenylphosphine group. The probe includes 4-bromomethylphenylboronic acid pinacol ester where borates converse to phenol upon binding to ONOO- resulting in increased quantum yield of fluorescence.
There is a linear relationship between the concentration of corresponding ROS and fluorescence intensity.
The authors tested the stability of fluorescent probe response in the various pH and found it relatively stable in the physiological range of pH changes. They also tested the selectivity of the probe to different ROS and found it highly selective to the ROS of interest.
To confirm that the probe can be used for biological applications, the authors performed a series of experiments with HeLa, RAW 264.7, and HepG 2 cells and observed high survival rates for the cells loaded with the probe in concentrations below 70 µM.
ROS/RNS generator SIN-1 was used to observe the probe's fluorescence increase as a reaction to extracellular ONOOˉ rise. ROS scavenger was used to observe how the probe's fluorescence attenuates with a decrease in extracellular ROS content.
The authors also tested the reaction of the probe to intracellular ONOO- changes and used Mito-Tracker to verify the colocalization probe with mitochondria in the cells.
To demonstrate the probe's capabilities as a two-photon tool, the zebrafish model was used for detecting ONOO- changes.
In my opinion, this work has great scientific merit. It demonstrates the development of a new fluorescent dye sensitive to a specific kind of ROS family. It shows comprehensive series of experiments to verify the capabilities of the newly developed tool. It is well enough constructed and written. The language can be a bit more polished, but I have no critical remarks, and in principle, the manuscript can be published as is.
Author Response
Dear Editor and Reviewers,
Thank you very much for your positive evaluation of this manuscript. According to your and the reviewer’s advice, the manuscript entitled “A two-photon fluorescent probe for visual detection of peroxynitrite in living cells and zebrafish” has been revised.
We highly appreciate the great efforts and significant guidance of the referees and your close study concerning our manuscript. These comments are of great importance for revising and improving our work. As you indicated, we have revised the manuscript seriously and carefully, as follows:
1) We have corrected some errors in the manuscript.
2) We have compared our probe with other published probes.
Sincerely yours,
Zhencai Xu
------------
Guanyun People’s Hospital, Lianyungang, Jiangsu, 222000, China
Zhencai Xu, Email: xu2454287273@163.com
------------
Response to the editor and reviewers’ comments
Independent Review Report, Reviewer 2
EVALUATION
In the manuscript, the authors claim to develop a novel ROS-oriented fluorescent probe which is specifically sensitive to ONOO- and can target mitochondria upon loading in the cells. Despite a variety of ROS detecting fluorescent probes available on the market now, new probes specific only to particular ROS of interest and targeting specific organelle presents a great interest to the research community.
The probe can freely permeate the cell's membrane and targets mitochondria due to the cationic triphenylphosphine group. The probe includes 4-bromomethylphenylboronic acid pinacol ester where borates converse to phenol upon binding to ONOO- resulting in increased quantum yield of fluorescence. There is a linear relationship between the concentration of corresponding ROS and fluorescence intensity. The authors tested the stability of fluorescent probe response in the various pH and found it relatively stable in the physiological range of pH changes. They also tested the selectivity of the probe to different ROS and found it highly selective to the ROS of interest. To confirm that the probe can be used for biological applications, the authors performed a series of experiments with HeLa, RAW 264.7, and HepG 2 cells and observed high survival rates for the cells loaded with the probe in concentrations below 70 µM. ROS/RNS generator SIN-1 was used to observe the probe's fluorescence increase as a reaction to extracellular ONOOˉ rise. ROS scavenger was used to observe how the probe's fluorescence attenuates with a decrease in extracellular ROS content. The authors also tested the reaction of the probe to intracellular ONOO- changes and used Mito-Tracker to verify the colocalization probe with mitochondria in the cells. To demonstrate the probe's capabilities as a two-photon tool, the zebrafish model was used for detecting ONOO- changes.
In my opinion, this work has great scientific merit. It demonstrates the development of a new fluorescent dye sensitive to a specific kind of ROS family. It shows comprehensive.
R: Thanks very much for your positive comment on our work, which has greatly encouraged us to further study. We have polished the whole manuscript to improve the language to make it better understood, and we will continue to work on the development of fluorescent probes in the future.
